# Polynomial-time quantum Gibbs sampling for the weak and strong coupling regime of the Fermi-Hubbard model at any temperature

Štěpán Šmíd [1] ✉, Richard Meister [1], Mario Berta [1,2] & Roberto Bondesan [1]

Quantum computers hold the potential to revolutionise the simulation of quantum many-body systems, with profound implications for fundamental physics and applications like molecular and material design. However, demonstrating quantum advantage in simulating quantum systems of practical relevance remains a significant challenge. In this work, we introduce a quantum algorithm for preparing Gibbs states of interacting fermions on a lattice with provable polynomial resource requirements. Our approach builds on recent progress in theoretical computer science that extends classical Markov chain Monte Carlo methods to the quantum domain. We derive a bound on the mixing time for quantum Gibbs state preparation by showing that the generator of the quantum Markovian evolution is gapped at any temperature up to a maximal interaction strength. This enables the efficient preparation of low-temperature states of weakly interacting fermions and the calculation of their free energy. We present exact numerical simulations for small system sizes that support our results and identify well-suited algorithmic choices for simulating the Fermi-Hubbard model beyond our rigorous guarantees.

Quantum computers promise to have a transformative impact on computing, as quantum algorithms are believed to solve certain computational problems significantly faster—potentially offering super-polynomial speed-ups over their classical counterparts. However, it is important to recognise that such substantial quantum advantages are far from generic. Among the most promising areas of application is the simulation of quantum many-body systems[1]. While quantum approaches to understanding the zero-temperature ground state physics of quantum many-body systems date back to the early days of quantum computing[2], recent progress in theoretical computer science has introduced new algorithmic techniques for preparing quantum Gibbs states at non-zero temperature. In particular, simulated Lindbladian evolution within the quantum circuit model has

emerged as a powerful tool for this task[3]. These so-called quantum Gibbs samplers represent the non-commutative analogues of the classically successful Markov chain Monte Carlo (MCMC) methods[4], and are expected to offer efficient access to non-zero temperature properties of quantum many-body systems, especially in regimes that are classically intractable.

Starting from the algorithmic breakthrough result[3], there is the physical intuition that quantum Gibbs samplers can perform well for some specific case instances relevant to computational physics and computational chemistry. This is contrasted to known worst-case hardness results[2,5–7], as well as to previously proposed intricate non-zero temperature quantum methods that are (partially) missing rigorous guarantees[8–14] or are believed to be computationally expensive

[1]Department of Computing, Imperial College London, London, UK. [2]Institute for Quantum Information, RWTH Aachen University, Aachen, Germany.
✉e-mail: s.smid23@imperial.ac.uk

on relevant finite-size instances for finite temperature[15–19]. The guiding idea behind the latest algorithmic Lindbladian constructions is to efficiently simulate thermalisation processes in nature, as, e.g., modelled by the Davies generator[20,21]. In particular, it is possible to bring together algorithmic efficiency with an exact notion of quantum detailed balance, and we refer to the recent quantum Gibbs sampler frameworks[22,23] as well as references therein for an extended discussion.

Classical MCMC algorithms are termed efficient when they converge to the Gibbs state in time polynomial or even logarithmic in system size[4]. In contrast, the recently proposed quantum Gibbs samplers based on algorithmic Lindbladian evolution are solely efficient in the sense that all algorithmic steps are implemented efficiently, but the complexity or overall runtime crucially relies on the so-called mixing time, which specifies for how long we need to conduct the evolution to get $\epsilon$-close to the Gibbs state. For an efficient algorithm, we would then require the Lindbladian to undergo fast mixing in time scaling polynomially with system size. This then has to be resolved for each physical system of interest on a case-by-case basis. The situation is akin to quantum adiabatic algorithms, where the runtime is governed by the Hamiltonian gap along the adiabatic path[24] and careful analytical studies are typically challenging.

With classical MCMC methods it is well-understood that rigorous bounds on the convergence time as studied in mathematical physics[25–29] often vastly overestimate the observed convergence times when running the algorithms in practice and reading out physical information, such as the partition function, relevant observables, or correlation functions. However, it is by design challenging to numerically run quantum Gibbs samplers to estimate practical mixing times—after all, we do not (yet) have reliable large-scale quantum computers and the classical simulation thereof is believed to be hard. As such, analytical bounds on mixing times are the first main tool to start our efficiency analysis. They would also serve as valuable benchmarks for future empirical studies of these models on fault-tolerant quantum computers, extending beyond the regimes with analytical guarantees.

In order to witness decisive quantum advantages, our goal is to provide quantum many-body systems with instances of quantum Gibbs samplers where

(i) Classical methods are not sufficient to conclusively determine the physics at non-zero temperature.
(ii) We can derive rigorous and efficient bounds on the mixing time, at least for certain non-trivial parameter regimes.
(iii) We can subsequently give informed heuristics on how to fine-tune quantum Gibbs samplers to see efficient mixing times for relevant specific case parameter regimes beyond analytical worst-case guarantees.

As an example, previous general results on efficient quantum mixing times have been recently obtained in the high-temperature limit[30,31], where, however, classical methods are proven to also be efficient[32]. Other problem-specific abstract results on efficient mixing time bounds from the theoretical computer science literature include random sparse Hamiltonians[33], random local Hamiltonians[34], the toric code[35], and parent Hamiltonians of shallow quantum circuits[36,37].

In contrast, here we focus on the quantum simulation of fermionic systems of practical relevance—specifically the Fermi-Hubbard model on $D$-dimensional lattices—and establish rigorous bounds on its corresponding mixing times in both the weakly and strongly interacting regimes. This model enjoys widespread applications in science, for example, to the Mott metal-insulator transition and to high-temperature superconductivity[38], while at the same time remaining challenging for classical methods[39,40]. As such, it further serves as a standard benchmark for computational methods[41], including quantum simulations using ultra-cold atoms[42]. There is also evidence that, in contrast to quantumly hard, glassy spin systems, interacting fermionic systems are computationally challenging due to their entanglement structure[43]—which is believed to be amenable to quantum methods (in regimes where classical ones might not be). Lastly, we emphasise again that the sought-after efficiency for specific cases and typical finite system sizes would not be in contradiction with the known worst-case hardness results for fermionic systems[6,7].

## Results

### Fermionic systems

We start with the recent algorithmic work[3], which derives that the mixing time of Lindbladian quantum Gibbs samplers can be upper bounded by giving lower bounds on the *spectral gap of the Lindbladian*. This spectral gap, in turn, is equal to the spectral gap of a corresponding *parent Hamiltonian*, which we analyse. Our approach is to first study free fermions, governed by the quadratic Hamiltonian

$$H = H_0 = \sum_{i,j=1}^{2n} \omega_i h_{ij} \omega_j , \quad \text{with } \{\omega_i, \omega_j\} = 2\delta_{ij} , \ \omega_i^\dagger = \omega_i ,$$

written in terms of Majorana fermions $\omega_i$. Note that the single-particle Hamiltonian $h$ includes the chemical potential terms, which effectively control the number of fermions in the system. Since our ensuing results allow for arbitrary chemical potentials, these methods can be used to study systems with desired levels of hole doping. We follow the general algorithmic quantum Gibbs sampler framework of ref. 22, which we review in detail in Supplementary Information, but the key point of the quantum algorithm is to simulate dynamics generated by an algorithmic Lindbladian of the form

$$\mathcal{L}^\dagger[\rho] = -i[G,\rho] + \sum_{a\in\mathcal{A}} \left( L_a \rho L_a^\dagger - \frac{1}{2}\{L_a^\dagger L_a, \rho\} \right) ,$$

which is fully specified after choosing a suitable set of jump operators $A^a$ and corresponding filter functions $f^a$. The Lindblad operators $L_a$ are then filtered operator Fourier transforms of the jump operators, while the coherent term $G$ is uniquely determined from the detailed-balance condition; this is the condition that guarantees the Gibbs state to be a fixed point of the Lindbladian evolution. Intuitively, these choices are akin to establishing the set of proposals and their acceptance probabilities in the Metropolis-Hastings algorithm. We make the specific design choice of Majorana jump operators, $\{A^a = \omega_a\}_{a=1}^{2n}$, and equal Gaussian filter functions $f^a = f$, obeying the required conditions. In particular, this allows us to *exactly* compute the finite spectral gap of the free fermionic parent Hamiltonian via Prosen's third quantisation formalism[44] as (see Methods)

$$\Delta_0 = 2 \cdot e^{-4\beta^2 \|h\|^2} \cosh(2\beta \| h \|) , \tag{1}$$

where $\beta = T^{-1}$ is the inverse temperature. This is lower bounded due to the locality of the system ensuring $\| h \| \le \mathcal{O}(1)$. On top of this, recognising that the dynamics is restricted to that of Gaussian states if we start from a Gaussian state, we calculate the covariance matrix of the evolved state, which, together with optimal trace norm bounds[45] allows us to prove a logarithmic upper bound on the mixing time for free fermions

$$t_{\text{mix}} \le \frac{1}{2\Delta_0} \log\left(\frac{2n}{\epsilon}\right) .$$

For the first time, this thus proves rapid mixing in logarithmic time of non-commuting Hamiltonians at any temperature.

Then, after observing that the free fermionic parent Hamiltonian in third quantisation itself simplifies to a (different) free fermionic Hamiltonian, we make use of Hastings' stability result[46–48] on the

spectral gap of free fermions under perturbation in order to quantitatively extend the finite spectral gap in the thermodynamic limit to the interacting parent Hamiltonian (Theorem 1). To lift the locality of interactions from the fermionic Hamiltonian to the Lindbladian's parent Hamiltonian, we use Lieb-Robinson bounds and employ matrix analysis methods and identities such as Duhamel's formula.

Our main finding is phrased most generally for lattice fermionic Hamiltonians with exponentially decaying interactions, henceforth termed quasi-local. Namely, we show that for such systems and at any constant temperature $T > 0$, there exists a constant maximal interaction strength below which the corresponding purified Gibbs states can be prepared efficiently. This result can be summarised in the following theorem and corollary:

**Theorem 1.** For any interacting quasi-local fermionic Hamiltonian $H = H_0 + \lambda V$ at any temperature $T > 0$, there exists a positive system-size-independent value $\lambda_{\max}$ such that the Lindbladian $\mathcal{L}^\dagger$ for Gibbs state preparation has a spectral gap $\Delta$ lower bounded by a constant for any $|\lambda| \le \lambda_{\max}$, closing at most linearly in $|\lambda|$ from that of the non-interacting case $H_0$.

**Corollary 1.1.** The mixing time of the Gibbs state preparation is upper bounded by $t_{\mathrm{mix}} = \mathcal{O}(n + \log(1/\epsilon))$, and the overall quantum complexity to create the quantum Gibbs state is $\widetilde{\mathcal{O}}(n^3 \mathrm{polylog}(1/\epsilon))$ and requires $\mathcal{O}(n)$ qubits; where $n$ denotes the system size and $\epsilon > 0$ the desired accuracy in trace norm.

Crucially, the constant on the interaction strength is independent of the system size, and thus we conclude that (weakly) interacting fermionic systems in fixed dimension and at any constant temperature can be efficiently simulated on quantum computers. Although free fermions are efficiently solvable, to the best of our knowledge, there is no provably efficient classical algorithm for the weakly-interacting regime, leading to a potential exponential quantum advantage. This is in contrast to the previously studied case of high temperatures[30,31], where they have shown stability around the infinite temperature limit, hence offering only a polynomial quantum advantage[32,49].

Our results apply, in particular, to the Fermi-Hubbard model. Its spinful version on a $D$-dimensional lattice is governed by the Hamiltonian

$$H_{\mathrm{FH}} = -t \sum_{\langle i,j \rangle, \sigma} \left( a_{i,\sigma}^\dagger a_{j,\sigma} + a_{j,\sigma}^\dagger a_{i,\sigma} \right) + U \sum_i a_{i,\uparrow}^\dagger a_{i,\uparrow} a_{i,\downarrow}^\dagger a_{i,\downarrow},$$

where $\langle \cdot, \cdot \rangle$ denotes neighbouring sites on the lattice, $\sigma \in \{\uparrow, \downarrow\}$ the spins, and $a_{i,\sigma}^{(\dagger)}$ the fermionic annihilation and creation operators on site $i$ with spin $\sigma$. The weakly-interacting limit corresponds to $U/t \lesssim 1$[39–41], which can then serve as an analytical starting point for further numerical investigations. Furthermore, while the Fermi-Hubbard model is exactly solvable for the $D = 1$ case[50], we emphasise that our results are equally valid in any dimension. We refer to ref. 39 for a recent review of exact and heuristic results for the Hubbard model. In particular, the state-of-the-art in powerful heuristic methods, such as tensor networks, allows one to study the ground state of the Fermi-Hubbard model up to lattices of size $16 \times 16$[51]. Notably, in two-dimensional systems with next-nearest-neighbour hopping, even the weak-coupling regime exhibits rich behaviour as the chemical potential is varied, with multiple competing instabilities emerging near the van Hove singularity[40]. Our algorithm enables the study of such phenomena with, in principle, arbitrarily high theoretical precision.

## Applications, extensions, and numerical simulations
As an application of Gibbs sampling, we adapt ref. 31, [Theorem 8] to calculate partition functions for the considered systems. Hence, this provides the means to resolve the physics in thermal equilibrium of the underlying quantum many-body system in an end-to-end fashion:

**Proposition 2.** For any interacting quasi-local fermionic Hamiltonian $H = H_0 + \lambda V$ at any temperature $T > 0$, there exists a positive system-size-independent value $\lambda_{\max}$ such that for any $|\lambda| \le \lambda_{\max}$ the corresponding partition function $Z$ is determined in quantum complexity $\widetilde{\mathcal{O}}(n^5 \epsilon^{-2})$, with success probability at least 3/4 and up to a relative error $\epsilon > 0$.

Further, our methods equally apply to the opposite regime $U/t \gg 1$ of the Fermi-Hubbard model. Here, we start by exactly solving the atomic limit $t = 0$ case of no hopping with the same choices of jump operators and filter functions (see Supplementary Information B3), after which we use adapted eigenvalue perturbation techniques to control finite $t > 0$. This result can be summarised in the following theorem:

**Theorem 3.** For any quasi-local perturbed atomic Hamiltonian $H = H_{\mathrm{atomic}} + \lambda V$ at any temperature $T > 0$, there exists a positive system-size-independent value $\lambda_{\max}$ such that the Lindbladian $\mathcal{L}^\dagger$ for Gibbs state preparation has a spectral gap $\Delta$ lower bounded by a constant for any $|\lambda| \le \lambda_{\max}$; closing at most as $\mathcal{O}(|\lambda|^\alpha)$ with arbitrary $\alpha < 1$ from that of the atomic case $H_{\mathrm{atomic}}$.

We again find that at any finite temperature, there exists a constant (system-size-independent) maximal $t$ below which the corresponding Gibbs states can be algorithmically created with the complexities as stated in Corollary 1.1. We emphasise that the flexibility of our proof techniques naturally lends itself to future explorations of other quantum many-body systems in various regimes, such as spin systems in external fields, where the spin-spin interaction is treated as the perturbation. Here, one has to take into account that provably efficient classical algorithms exist for quantum perturbations of atomic Hamiltonians also at low temperatures[52].

We perform small-scale exact classical simulations for the weakly-interacting spinless and spinful Fermi-Hubbard model in order to trial the hidden asymptotic constants in our analytical result. We find reasonable finite-gap behaviour and confirm, in particular, the predicted system size-independent scaling. Next, we aim to heuristically improve the algorithmic choices in order to shorten the mixing times beyond our theoretical guarantees. In general, the temperature dependence scales unfavourably in our analytical result, and this becomes at first equally visible in the numerical analysis. However, varying the choice of the jump operators from Majoranas to Paulis and the filter functions from Gaussian to Metropolis (see Supplementary Information Equation (A8)), we observe a spectral gap of the Lindbladian whose magnitude is no longer decreasing when lowering the temperature for the small system sizes we simulate. This would make the overall algorithmic dependence on the inverse temperature $\beta$ only quasi-quadratic, significantly broadening the practical applicability of the algorithm.

To probe beyond the weakly-interacting regime covered by our analytical bounds, we investigate intermediate coupling strengths with $2 \lesssim U/t \lesssim 6$ for spinless $D = 1, 2$ systems at different temperatures. Our numerical results suggest that, for one-dimensional and nearly one-dimensional cases, the algorithm remains efficient even at arbitrary coupling strength, incurring only a small additional cost that scales poly-logarithmically with $U/t$. For higher dimensions, however, the diversity of physical phenomena exceeds the scope of our simulations, currently preventing any definitive conclusions beyond our theoretical analysis. Details are provided in Figs. 1 and 2, and the corresponding simulation code is available at ref. 53.

## Discussion
We have shown for the first time that the Gibbs states of weakly-interacting fermions in any dimension and at any constant

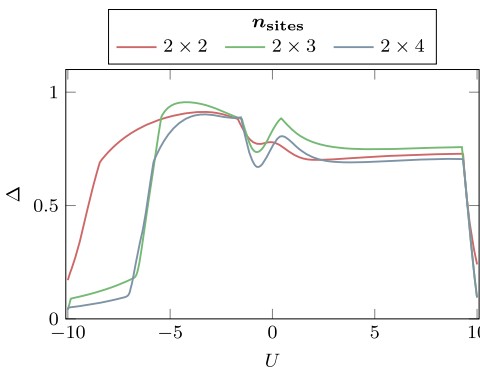

**Fig. 1 | Numerical results beyond our analytical guarantees.** Plotting the gap $\Delta$ of the full Lindbladian $\mathcal{L}^\dagger$ associated with the spinless $D = 2$ Fermi-Hubbard thermal state with design choices beyond our analytical results—when using the Metropolis filter function and single site Pauli jump operators instead—as a function of the coupling strength $U$. Here we plot different system sizes separately, for the case $\beta = 1$ and $t = 1$, demonstrating a large spectral gap in the regime of intermediate coupling $2 \lesssim U/t \lesssim 6$, which also does not seem to close with growing inverse temperature $\beta$ (see Supplementary Figs. 7 and 9). At what coupling strength the gap closes is controlled by the support of the filter function, incurring only poly-logarithmic additional algorithmic cost in $U/t$.

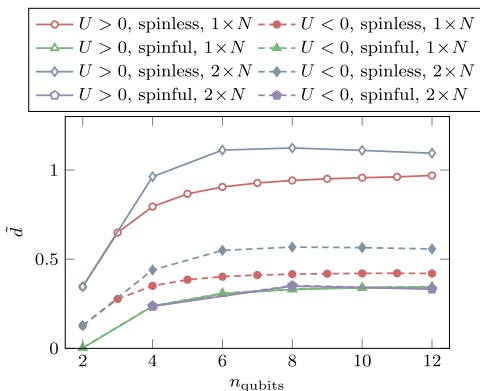

**Fig. 2 | Numerical results confirming our analytical findings.** Plotting the slope $\bar{d} = \mp \frac{\partial \Delta}{\partial U}\big|_{U=0^\pm}$ under which the spectral gap $\Delta$ of the full Lindbladian $\mathcal{L}^\dagger$ closes from that of the unperturbed Lindbladian $\mathcal{L}_0^\dagger$ in the analytically bounded regime, as the system size $n$ grows, at $\beta = 1$. As per our main result (Theorem 1), leading to the complexities as stated in Corollary 1.1, this quantity has to be upper-bounded uniformly in $n$. We refer to Supplementary Figs. for data for more sets of parameters exhibiting different types of behaviours.

temperature can be efficiently prepared in polynomial time on quantum computers. We have further demonstrated that this allows us to directly determine physical properties of the underlying systems—such as the partition function—in an end-to-end fashion while staying in polynomial quantum complexity. This is in contrast to, e.g., phase estimation-based quantum methods, which suffer from the so-called state preparation complexity bottleneck that generically hides large complexity overheads[1,41]. As such, we believe that the presented methods have great potential for resolving the relevant non-zero-temperature physics of the Fermi-Hubbard and other fermionic models in classically challenging regimes, hence shedding light on the unknown parts of their phase diagrams. Other applications of efficient quantum Gibbs samplers to be explored are in quantum approaches to optimisation[54,55] and quantum machine learning[56].

Going forward, it would be interesting to improve the exact dependencies on all the relevant parameters in our main result, as well as to work out all the hidden constants therein. One might then also fine-tune the choice of jump operators and filter functions for specific

systems and parameter regimes, hopefully even improving on the current cubic complexity in system size, potentially all the way down to quasi-linear. The rapid mixing of free fermions serves as a good indication for the weakly-interacting case to also mix rapidly, which would already bring down one factor of $n$. As we have seen, the cubic dependency on the system size is then mainly due to the quadratic dependence on the number of jump operators taken, which needs to be linear so that the Lindbladian dynamics can be irreducible and ergodic. However, we could potentially use different sets of jump operators at different times, consisting of as few as a single jump operator at any given time. The generated dynamics then could not be described by a single quantum Markov semigroup, and any one of the corresponding Lindbladians would not be able to get us from an arbitrary starting position to the Gibbs state; however, the combination of all of them could create a complicated path in the state space eventually getting us to the Gibbs state, with potentially significantly better dependency on the system size.

In order to determine the classical-quantum efficiency boundary for the Fermi-Hubbard model in the absence of reasonable quantum computers, it would be important to perform larger-scale classical simulations of the quantum Gibbs samplers, with varying design parameters to estimate the relevant spectral gap (e.g., based on tensor network methods[51,57,58]). Especially for translationally invariant systems in $D = 1$, for which the parent Hamiltonian would also inherit the invariance, one could use the imaginary-time iTEBD algorithm to simulate the evolution or iDMRG to calculate the spectral gap for infinite-sized systems. Any such findings should then be compared to the state-of-the-art classical results[39,40] to make statements about large quantum advantages for practically relevant problems.

Finally, our presented proof methods based on eigenvalue perturbation techniques also seem promising to explore other quantum many-body systems in different regimes, including, for example, bosonic systems. As a first step, the extensions presented in Supplementary Information B3 are easily shown to hold for any Hamiltonians that are separable in the lattice sites.

## Methods
### Free fermions
Let's consider a free fermionic Hamiltonian $H_0 = \boldsymbol{\omega}^T \cdot h \cdot \boldsymbol{\omega} = \sum_{i,j=1}^{2n} \omega_i h_{ij} \omega_j$ given in terms of Majorana fermions, with $h$ being a Hermitian and anti-symmetric matrix. We denote these operators using bold matrix-vector notation for convenience. We start by choosing the jump operators in the algorithmic Lindbladian to be Majorana fermions, $\{A^a = \omega_a\}_{a=1}^{2n}$, and the filter functions $f^a$ to be equal and real in the Fourier space. Note that the conditions on the filter function then say that $\hat{f}^a(\nu) = \hat{f}(\nu) = q(\nu) \cdot e^{-\beta\nu/4}$ with $q(\nu)$ real and even.

We readily find the Heisenberg time-evolved jump operators as

$$\mathbf{A}(t) = e^{iH_0 t} \mathbf{A} e^{-iH_0 t} = e^{iH_0 t} \boldsymbol{\omega} e^{-iH_0 t} = e^{-4iht} \cdot \boldsymbol{\omega},$$

and the Lindblad operators are then just

$$\mathbf{L} = \int_{-\infty}^{\infty} f(t) e^{-4iht} \, \mathrm{d}t \cdot \boldsymbol{\omega} = \hat{f}(-4h) \cdot \boldsymbol{\omega}.$$

Hence we find that

$$\sum_{a \in \mathcal{A}} L_a^\dagger L_a = \boldsymbol{\omega}^T \cdot q(4h)^2 \cdot \sinh(2\beta h) \cdot \boldsymbol{\omega} + \mathrm{Tr}\left(\hat{f}(-4h)^2\right),$$

but since $[\boldsymbol{\omega}^T \cdot B \cdot \boldsymbol{\omega}, \boldsymbol{\omega}^T \cdot C \cdot \boldsymbol{\omega}] = \boldsymbol{\omega}^T \cdot [B, C] \cdot \boldsymbol{\omega}$, this sum will commute with the Hamiltonian $H_0$. This, in turn, means it is invariant under time

evolution, and so the coherent term $G$ will be proportional to $\hat{g}(0)$, as

$$G = \int_{-\infty}^{\infty} g(t) \cdot \sum_{a \in \mathcal{A}} L_a^\dagger L_a \, dt \propto \hat{g}(0).$$

But since $\hat{g}(\nu) \propto \tanh(\beta\nu/4)$, this means that the coherent term vanishes, $G = 0$.

For future convenience, before analysing the spectrum of the Lindbladian, let's consider the similarity transformation $\mathcal{H}_0[\rho] = \sigma_\beta^{-1/4} \cdot \mathcal{L}_0^\dagger[\sigma_\beta^{1/4} \cdot \rho \cdot \sigma_\beta^{1/4}] \cdot \sigma_\beta^{-1/4}$ into the parent Hamiltonian. This superoperator is Hermitian due to the quantum detailed-balance condition. As this is a similarity transformation, the spectrum of $\mathcal{H}_0$ will be the same as of $\mathcal{L}_0^\dagger$. We can calculate that

$$\sigma_\beta^{-1/4} \mathbf{L} \sigma_\beta^{1/4} = \hat{f}(-4h) \cdot e^{-\beta h} \cdot \boldsymbol{\omega},$$

and the QDB condition also ensures $\sigma_\beta^{-1/4} L_a \sigma_\beta^{1/4} = \sigma_\beta^{1/4} L_a^\dagger \sigma_\beta^{-1/4}$. Bringing these calculations together and following Prosen's third quantisation[44], which we review in Supplementary Section A3, the parent Hamiltonian simplifies to

$$\mathcal{H}_0 \cong -\mathbf{c}^\dagger \cdot S \cdot \mathbf{c} + \mathbf{c} \cdot S \cdot \mathbf{c}^\dagger + \mathbf{c}^\dagger \cdot A \cdot \mathbf{c}^\dagger + \mathbf{c} \cdot A \cdot \mathbf{c} - \mathrm{Tr}\left(\sqrt{S^2 + A^2}\right).$$

Here we have restricted the Hilbert space to that of physical states with even numbers of Majorana fermions; and $S = q(4h)^2$, $A = q(4h)^2 \sinh(2\beta h)$, and $\{c_i^\dagger, c_i\}_{i=1}^{2n}$ is a set of $2n$ canonical fermionic creation and annihilation operators. This is just a quadratic fermionic operator, and hence its complete spectrum, which is the same as of $\mathcal{L}_0^\dagger$, is straightforwardly calculable as

$$\mathrm{spec}(\mathcal{L}_0^\dagger) = \left\{ \sum_{i=1}^{2n} (-1 + (-1)^{x_i}) \cdot q(4\epsilon_i)^2 \cosh(2\beta\epsilon_i) \right\}_{x \in \{0,1\}^{2n}},$$

where $\epsilon_i \in \mathrm{spec}(h)$. In particular, the corresponding spectral gap between the highest and second-highest eigenvalue is

$$\Delta_0 = 2 \cdot \min_i q(4\epsilon_i)^2 \cosh(2\beta\epsilon_i).$$

Taking the Gaussian filter function, $q(\nu) = e^{-\beta^2\nu^2/8}$, the spectral gap would be monotonically decaying with $\|h\|$, and hence lower bounded by a constant for local systems obeying $\|h\| \leq \mathcal{O}(1)$. This argument also assures that the Gibbs state is the unique fixed point of the dynamics generated by $\mathcal{L}_0^\dagger$.

On top of this, by recognising that when we start with a Gaussian state $\rho_0$ and evolve it with a quadratic Lindbladian, we will stay within the subspace of Gaussian states, we can restrict our view to the evolution of the covariance matrix $\Gamma_{ij}(t) = \frac{i}{2} \mathrm{Tr}([\omega_i, \omega_j]\rho(t))$. Denoting the initial covariance matrix by $\Gamma_0$, we can straightforwardly solve its equation of motion[59] and get that $\Gamma(t) = \frac{i}{2}\tanh(2\beta h) + e^{-2q(4h)^2\cosh(2\beta h)\cdot t} \cdot (\Gamma_0 - \frac{i}{2}\tanh(2\beta h)) \cdot e^{-2q(4h)^2\cosh(2\beta h)\cdot t}$. We can also check that the covariance matrix of the Gibbs state $\sigma_\beta$ is $\Gamma_{\sigma_\beta} = \frac{i}{2}\tanh(2\beta h) = \Gamma(\infty)$, and so the evolution indeed converges to the Gibbs state.

Finally, we can use optimal trace norm bounds obtained in ref. 45, which tell us that $\left\|\rho(t) - \sigma_\beta\right\|_{\mathrm{Tr}} \leq \frac{1}{2}\|\Gamma(t) - \Gamma_{\sigma_\beta}\|_{\mathrm{Tr}}$, which we can set smaller to $\epsilon$ and solve for $t$ and hence deduce that

$$t_{\mathrm{mix}} \leq \frac{1}{4\min_i q(4\epsilon_i)^2 \cosh(2\beta\epsilon_i)} \log\left(\frac{2n}{\epsilon}\right).$$

## Locality of parent Hamiltonians

Ref. 22 shows that if $H$ is a geometrically local Hamiltonian, $A_a$ are local, and the filter function is Gaussian, then the Lindblad operators $L_a$ are quasi-local and $G$ is a sum of quasi-local terms. Here, we extend this result and discuss the locality properties of the parent Hamiltonian for fermionic systems and systems with exponentially decaying interactions. The quasi-locality of the parent Hamiltonian will be an important ingredient in the proofs of gap stability we present below.

We shall again consider the (general) transformation into the parent Hamiltonian

$$\mathcal{H}[\rho] = \sigma_\beta^{-1/4} \cdot \mathcal{L}^\dagger\left[\sigma_\beta^{1/4} \cdot \rho \cdot \sigma_\beta^{1/4}\right] \cdot \sigma_\beta^{-1/4}, \quad (4.1)$$

and also define the transformed operators appearing therein by $\tilde{L}_a = \sigma_\beta^{-1/4} L_a \sigma_\beta^{1/4}$ and $\tilde{G} = \sigma_\beta^{-1/4} G \sigma_\beta^{1/4}$.

**Proposition 4.** Consider a Hamiltonian $H$ with interactions that decay at least exponentially, and local jump operators $A^a$ with Gaussian filter functions. Then the parent Hamiltonian (4.1) is a sum of quasi-local terms.

**Proof.** We define local approximations of $\tilde{L}_a$ and $\tilde{G}$ using

$$\tilde{L}_a^{(r)} = \int_{-\infty}^{\infty} f^a(t + i\beta/4) e^{iH_{B_r(a)}t} A^a e^{-iH_{B_r(a)}t} \, dt,$$

$$\tilde{G}_a^{(r)} = \int_{-\infty}^{\infty} g(t + i\beta/4) e^{iH_{B_r(a)}t} \left(L_a^{(r)\dagger} L_a^{(r)}\right) e^{-iH_{B_r(a)}t} \, dt,$$

where $B_r(a)$ is a ball of radius $r$ around the support of $A^a$ and $H_\Omega = \sum_{I|I\cap\Omega\neq\emptyset} h_I$ is the truncated Hamiltonian to the region $\Omega$. The conjugation by the Gibbs state translates into the shift of the functional arguments in the integrands, and is explained in Supplementary Information A2, D.

Here we shall use a weaker version of the Lieb-Robinson bound than the one for local systems from ref. 60 [Lemma 5] used in ref. 22 [Prop. 20], which also holds for exponentially decaying Hamiltonian interactions, and tells us that

$$\left\|e^{iHt}A^a e^{-iHt} - e^{iH_{B_r(a)}t}A^a e^{-iH_{B_r(a)}t}\right\|$$
$$\leq \|A^a\| \min\{2, Je^{-\mu r}(e^{\mu v|t|} - 1)\} \quad (4.2)$$

for some constants $J$, $v$, and $\mu$. From now on, we shall assume $\|A^a\| \leq 1$. Using the Gaussian filter, it will follow that

$$\|\tilde{L}_a - \tilde{L}_a^{(r)}\| \leq Ce^{-\mu r},$$
$$\|\tilde{G}_a - \tilde{G}_a^{(r)}\| \leq \tilde{C}e^{-\bar{\mu}r};$$

calculations of which are detailed in the Supplementary Information A4.

Note that the Lieb-Robinson bound, which follows from the bound on the commutator with the Hamiltonian terms, is true in our fermionic setting independently of whether $A_a$ is even or odd in the number of fermions, since the constituent Hamiltonian terms are always even, so that (4.2) still holds. We refer to ref. 61 for more on Lieb-Robinson bounds and locality for fermions.

## Bounds on the strength of the perturbation

In this section, we shall think of the parent Hamiltonian $\mathcal{H}$ of the interacting system as a perturbation of the free fermionic parent Hamiltonian $\mathcal{H}_0$. The expectation is that for a small enough interaction strength $\lambda$ of the system's Hamiltonian $H = H_0 + \lambda V$, the spectral gap of

the parent Hamiltonian $\mathcal{H}$ should remain constant. To prove this idea, we will make use of results about stability of spectral gap under perturbation for free fermions[46–48], but before that we would need to understand how the strength of the parent Hamiltonian perturbation $\mathcal{V} = \mathcal{H} - \mathcal{H}_0$ depends on $\lambda$.

We start by deriving the following Lemma from Duhamel's formula (see Supplementary Information D):

**Lemma 5**. For any operator $O$, Hermitian operators $H_0$, $V$, and $\lambda, \alpha \in \mathbb{C}$, we have

$$\left\| e^{\alpha(H_0 + \lambda V)} O e^{-\alpha(H_0 + \lambda V)} - e^{\alpha H_0} O e^{-\alpha H_0} \right\|$$
$$\leq |\lambda| \, |\alpha| \max_{s \in [0,1]} \left\| \left[ V, e^{s\alpha H_0} O e^{-s\alpha H_0} \right] \right\|.$$

As previously, we shall denote the operators appearing in the full parent Hamiltonian by $\tilde{L}_a$ and $\tilde{G}$, and we shall also denote their corresponding counterparts appearing in the free fermionic parent Hamiltonian by $\tilde{L}_a^0$ and $\tilde{G}^0$. Then to understand the strength of $\mathcal{V} = \mathcal{H} - \mathcal{H}_0$, we will need to bound $\| \tilde{L}_a - \tilde{L}_a^0 \|$ and $\| \tilde{G}_a - \tilde{G}_a^0 \|$.

Using our Lemma together with the exact solution for time evolution in the free fermionic case, we find that

$$\left\| e^{H(\beta/4 + it)} A^a e^{-H(\beta/4 + it)} - e^{H_0(\beta/4 + it)} A^a e^{H_0(\beta/4 + it)} \right\|$$
$$\leq c_1 |\lambda| \cdot |\beta/4 + it| \cdot e^{c_2|\beta/4 + it|},$$

where we have assumed that $\| h \|_\infty \leq \mathcal{O}(1)$ and that the Hamiltonian contains only terms with even numbers of fermions. Importantly, the constants appearing in this expression are independent of the system size.

Hence, it immediately follows that

$$\| \tilde{L}_a - \tilde{L}_a^0 \| \leq |\lambda| \int_{-\infty}^{\infty} |f^a(t)| c_1 |\beta/4 + it| e^{c_2|\beta/4 + it|} \mathrm{d}t = c_3 |\lambda|,$$

for some constant $c_3$ independent of the system size. As the dissipative part of the perturbation $\mathcal{V}$ is a sum over different products or tensor products of this or equivalent expressions, we can conclude that the strength of this part grows linearly in $|\lambda|$, independently of the system size (see Supplementary Information B2 for details).

Now looking at the coherent term, we shall split it up into quasi-local contributions $G = \sum_a G_a$ with $G_a = \int_{-\infty}^{\infty} g(t) e^{iHt} L_a^\dagger L_a e^{-iHt} \mathrm{d}t$. Then we similarly need to bound $\| \tilde{G}_a - \tilde{G}_a^0 \|$, which we will split into a part depending on $\left\| L_a^\dagger L_a - L_a^{0\dagger} L_a^0 \right\|$ and a part depending on

$$\left\| e^{H(\beta/4 + it)} L_a^{0\dagger} L_a^0 e^{-H(\beta/4 + it)} - e^{H_0(\beta/4 + it)} L_a^{0\dagger} L_a^0 e^{-H_0(\beta/4 + it)} \right\|.$$

Here we can again use our Lemma to bound this second contribution by

$$|\lambda| |\beta/4 + it| \max_{s \in [0,1]} \left\| \left[ V, e^{s(\beta/4 + it)H_0} L_a^{0\dagger} L_a^0 e^{-s(\beta/4 + it)H_0} \right] \right\|.$$

Using the exact solution $L_a^0 = \sum_i \hat{f}(-4h)_{ai} \omega_i$, we can upper-bound this further like

$$\max_{s \in [0,1]} \left\| \left[ V, e^{s(\beta/4 + it)H_0} L_a^{0\dagger} L_a^0 e^{-s(\beta/4 + it)H_0} \right] \right\|$$
$$\leq 2c_2 e^{c_3 \beta/4} \cdot w_h(t) \cdot \| \hat{f}(-4h) \|_\infty^2,$$

where $w_h(t)$ is system-size-independent function growing subexponentially in $t$ (as discussed in Supplementary Information E). Observe that the previous argument for bounding the conjugated

expression $\left\| \sigma_\beta^{-1/4} L_a^\dagger L_a \sigma_\beta^{1/4} - \sigma_{\beta,0}^{-1/4} L_a^{0\dagger} L_a^0 \sigma_{\beta,0}^{1/4} \right\|$ also shows that $\left\| L_a^\dagger L_a - L_a^{0\dagger} L_a^0 \right\| \leq c_4 |\lambda|$. Finally, this means that

$$\| \tilde{G}_a - \tilde{G}_a^0 \|$$
$$\leq \int_{-\infty}^{\infty} |g(t)| \cdot \left( c_4 |\lambda| + |\lambda| |\beta/4 + it| c_5 w_h(t) \right) \mathrm{d}t = c_6 |\lambda|,$$

where the convergence is ensured by the decay bounds of $g(t)$ obtained in ref. 22 [Lemma 30].

This proves that the strength of the perturbation of the parent Hamiltonian is upper bounded by a constant multiple of the strength of the perturbation of the system's Hamiltonian, uniformly in system size, i.e., that $\mathcal{V}_a$, where $\mathcal{V} = \sum_{a \in \mathcal{A}} \mathcal{V}_a$, is upper bounded like $\| \mathcal{V}_a \| \leq c|\lambda|$. To match the locality definition we will need to use exactly, we would further make a standard argument by writing this perturbation as a telescoping sum over different radii (see Supplementary Information B2).

In Supplementary Lemma E.1, we also present a slightly weaker notion of this result, with the strength bounded by $|\lambda|^\alpha$ for an arbitrary constant $\alpha < 1$ for small enough $|\lambda|$, which works for general Hamiltonians, and hence can be used to obtain our secondary result for perturbations around the atomic limit.

## Constant gap and fast mixing

In our previous sections, we have proven that for a quasi-local interacting fermionic Hamiltonian $H = H_0 + \lambda V$ and with our algorithmic choices, the parent Hamiltonian $\mathcal{H}_0$ has $[J, \nu]$-decay and the perturbation $\mathcal{V}$ has $(c|\lambda|, \mu)$-decay as per Definitions 1 and 2 of ref. 46 (see Supplementary Information B2 for review). Hence, we can finish our discussion of the stability of the spectral gap by using [ref. 46, Corollary 1] to show that the gap of $\mathcal{H}$ closes at most linearly in $|\lambda|$ from that of $\mathcal{H}_0$ – uniformly in system size–and hence is lower bounded by a constant as per our main Theorem 1.

Finally, this result allows us to bound the mixing time $t_{\mathrm{mix}}$ of the Lindbladian $\mathcal{L}^\dagger$, defined as the minimal time necessary to get us $\epsilon$-close to the Gibbs state from an arbitrary initial position, and show *fast* mixing in polynomial time bounded by

$$t_{\mathrm{mix}} \leq \frac{\log\left( \frac{2}{\epsilon} \| \sigma_\beta^{-1/2} \| \right)}{\Delta} = \mathcal{O}(n + \log(1/\epsilon)).$$

Together with the complexity analysis from ref. 22 [Theorem 34], this gives us the overall time complexity of the algorithm as stated in Corollary 1.1.

## Calculating the partition function

As a possible application of the efficient Gibbs state preparation, we adapt the strategy from ref. 31 for calculating partition functions $Z_\beta(\lambda_i) = \mathrm{Tr}\,(e^{-\beta H(\lambda_i)})$ to the case of interacting fermionic systems, where we have denoted $H(\lambda_i) = H_0 + \lambda_i V$. Note that we can calculate the non-interacting partition function explicitly as

$$Z_\beta(0) = \prod_{i=1}^n 2 \cosh(2\beta \epsilon_i),$$

where the product is taken only over one $\epsilon_i \in \mathrm{spec}(h)$ from each symplectic pair $\pm \epsilon_i$. By measuring the observable $e^{\beta H(\lambda_i)} e^{-\beta H(\lambda_{i+1})}$ in the state $\sigma_\beta(\lambda_i)$, we would obtain the ratio $\frac{Z_\beta(\lambda_{i+1})}{Z_\beta(\lambda_i)}$. Preparing this observable and the Gibbs state will require access to block encodings of $H_0$ and $V$, from which we get a block encoding for $H(\lambda_i)$ via LCU, and hence a block encoding for the observable and the Hamiltonian simulation via QSVT. By choosing a schedule $0 = t_1 \leq t_2 \leq \cdots \leq t_{l-1} \leq t_l = |\lambda|$ and

denoting $\lambda_i = t_i \frac{\lambda}{|\lambda|}$, we can calculate $Z_\beta(\lambda)$ as a telescoping product

$$Z_\beta(\lambda) = Z_\beta(0) \prod_{i=1}^{l-1} \frac{Z_\beta(\lambda_{i+1})}{Z_\beta(\lambda_i)}.$$

We refer to ref. 31 [Appendix C] for the details of these calculations, the gist of which lies in choosing the schedule such that $t_{i+1} - t_i = \Theta(n^{-1})$, and so $l = \Theta(n)$ as $\lambda = \Theta(1)$. Then we would prepare the Gibbs states $\sigma_\beta(\lambda_i)$ and measure the expectation values of the observables $e^{\beta H(\lambda_i)} e^{-\beta H(\lambda_{i+1})}$ for each $i \in [l-1]$ at least $\Theta(n\epsilon^{-2})$ times. Averaging these measurements and calculating the partition function using the telescoping product would yield the result in Proposition 2.

## Data availability
We have not analysed any datasets as our work proceeds within a theoretical and mathematical approach. The figures shown contain all the available data, which was generated using the provided code.

## Code availability
Source code for simulating the spectral gap is available at https://github.com/Quantum-AI-Lab-ICL/Quantum-Gibbs-Sampling.

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

## Acknowledgements

During the write-up of our manuscript, we have become aware that Yu Tong and Yongtao Zhan were independently working on the same topic[62]. We thank them for agreeing on a date for uploading to the arXiv together. All authors acknowledge support from the EPSRC Grant number EP/W032643/1. M.B. acknowledges funding by the European Research Council (ERC grant agreement no. 948139) and the Excellence Cluster Matter and Light for Quantum Computing (ML4Q). The authors acknowledge the use of the Imperial College Research Computing Service[63] to obtain some of the results in this work.

## Author contributions

The initial directions were conceptualised by R.B., M.B., and Š.Š. The main calculations and proofs were developed by Š.Š. and R.B., with Š.Š. focusing on weakly-interacting fermions and R.B. on the atomic limit. R.M. and Š.Š. developed the numerical simulations. All authors contributed to conceptual discussions and the write-up of the manuscript.

## Competing interests

The authors declare no competing interests.
