## [Transparent Peer Review file · Nature Communications]

Polynomial-Time Quantum Gibbs Sampling for the Weak and Strong Coupling Regime of the Fermi-Hubbard Model at any Temperature

Corresponding Author: Mr Štěpán Šmíd

Version 0:

Reviewer comments:

Reviewer #2

(Remarks to the Author)

I have already reviewed this manuscript for [journal name redacted].

For the overall summary, as well as the strengths and weaknesses of the work, I refer the editor and authors to that report.

Here, I focus on the revised version and on points specific to Nature Communications.

The authors have substantially improved the manuscript.

The rigorous scope (weak coupling with a system-size, independent λ_{\max}), the free-fermion rapid mixing result, and the numerical evidence for wider regimes are now clearly separated. The presentation has become much clearer, and the paper has the potential to make a strong impact.

Before recommending acceptance, I believe the following changes are essential:

i) Title precision

The current title may mislead readers into thinking the results apply to the full Fermi-Hubbard model. The title must explicitly reflect the rigorous scope.

For instance "Polynomial-Time Quantum Gibbs Sampling for the Weak-Coupling Regime of the Fermi-Hubbard Model at Any Temperature"

ii) Statement of λ_{\max} .

In Theorem II.1, please explicitly state that the threshold λ_{\max} is independent of system size and temperature. This property is described in the text, but it should also be included in the theorem statement itself for clarity.

iii) Terminology consistency ("fast" and "rapid" mixing).

The manuscript uses the term fast mixing (Sec.~V.D) without a formal definition, while rapid mixing is explicitly defined (logarithmic-time mixing) for free fermions.

To avoid confusion, the authors should either

(a) unify the terminology and only use "rapid mixing" with an explicit definition, or

(b) provide a precise definition of "fast mixing" (linear-time mixing) and use both terms consistently.

With these clarifications, I am glad to recommend acceptance of the manuscript.

(Remarks on code availability)

Dear Reviewers,

We thank you for considering our manuscript. We have now implemented all the suggested changes. In particular, on top of previously discussed points, we have

- appended the title with the explicit qualifier specifying in which regimes do our theoretical guarantees hold,
- clarified the dependencies of λ_{\max} in the statements of the theorems,
- added definition of fast mixing at two points in the main text, as well as a full technical definition in the Background section in Supplementary Information, properly distinguishing between fast and rapid mixing,

as well as some other minor stylistic improvements. Further, we noticed that the rapid mixing result for free fermions straightforwardly generalises from the initial state being the maximally mixed one to an arbitrary convex combination of Gaussian states, and so we've included that for the reader's convenience.

Thank you for your time spent reviewing our work, and we hope you find our responses satisfactory.

Sincerely,

Štěpán Šmíd, Richard Meister, Mario Berta, and Roberto Bondesan